# Three-Step Synthesis of the Antiepileptic Drug Candidate Pynegabine

**DOI:** 10.3390/molecules28134888

**Published:** 2023-06-21

**Authors:** Yi-Jing Sun, Ya-Ling Gong, Shi-Chao Lu, Shi-Peng Zhang, Shu Xu

**Affiliations:** 1State Key Laboratory of Bioactive Substance and Function of Natural Medicines and Beijing Key Laboratory of Active Substance Discovery and Druggability Evaluation, Institute of Materia Medica, Chinese Academy of Medical Sciences and Peking Union Medical College, 2A Nanwei Road, Xicheng District, Beijing 100050, China; sunyijing2021@163.com (Y.-J.S.); ylgong@imm.ac.cn (Y.-L.G.); lushichao@imm.ac.cn (S.-C.L.); 2College of Chemical and Pharmaceutical Engineering, Hebei University of Science and Technology, Shijiazhuang 050018, China

**Keywords:** antiepileptic drug, pynegabine, Buchwald–Hartwig cross coupling

## Abstract

Pynegabine, an antiepileptic drug candidate in phase I clinical trials, is a structural analog of the marketed drug retigabine with improved chemical stability, strong efficacy, and a better safety margin. The reported shortest synthetic route for pynegabine contains six steps and involves the manipulation of highly toxic methyl chloroformate and dangerous hydrogen gas. To improve the feasibility of drug production, we developed a concise, three-step process using unconventional methoxycarbonylation and highly efficient Buchwald–Hartwig cross coupling. The new synthetic route generated pynegabine at the decagram scale without column chromatographic purification and avoided the dangerous manipulation of hazardous reagents.

## 1. Introduction

Epilepsy, a chronic disorder of the brain, is one of the most common neurological diseases, affecting around 50 million people globally [1]. Clinical antiepileptic drugs (AEDs) are only effective in about 70% of patients with epilepsy, making the discovery of new AEDs with novel mechanisms and/or low side effects critically important [2,3]. Retigabine (Figure 1), an AED approved by the U.S. Food and Drug Administration (FDA) in 2011, is the first and only marketed drug functioning as an agonist of KCNQ2 voltage-gated potassium ion channels [4]. However, in 2013, the FDA added warnings to the drug label emphasizing the serious risks associated with the use of retigabine, which included skin discoloration and retinal pigment abnormalities [5]. Subsequently, the production of retigabine was permanently discontinued in 2017 due to the declining market for the medication [6]. Several pigmented dimers of retigabine have been identified in an in vivo study, suggesting the correlation between the discoloration and the triaminobenzene segment of retigabine [7]. Therefore, the above adverse events of retigabine seem to originate from its chemical structure but not from the KCNQ2 target [8]. Accordingly, great efforts are made to search for new KCNQ2 agonists for AED candidates based on the optimization of retigabine structure [9,10,11,12,13,14,15,16].

Pynegabine, also known as HN37, is a structural analog of retigabine (Figure 1) [17]. Its ratios of the brain to blood exposure are 10 times more than those of retigabine in mice and rats. The EC_50_ of pynegabine for KCNQ2 channels and KCNQ2/KCNQ3 channels are also 55- and 125-fold more potent than those of retigabine. In a maximal electroshock seizure model test, the TD_50_/ED_50_ ratio of pynegabine was at least eight-fold bigger than those of retigabine and two other AEDs (levetiracetam and topiramate) which show a better safety margin. More importantly, by change of the triaminobenzene segment in retigabine, pynegabine exhibited much better stability for temperature, humidity, and light, implying a lower risk of discoloration [18]. Pynegabine has progressed into phase I clinical trials for epilepsy treatment in China (ChinaDrugTrials Identifier: CTR20201676, CTR20222616).

There are two synthetic routes reported for pynegabine, both starting from 2,6-dimethylaniline (**1**, Figure 1). The first-generation route prepared pynegabine at the milligram scale with a 30% total yield in 10 steps, out of which four steps applied column chromatography for purification [17,18]. Since this route is not feasible for large-scale production, a second-generation route was developed, achieving the synthesis of 10 g of pynegabine with a 30% total yield over six steps [17]. Although the latter synthetic route eliminated the unfavorable chromatographic purification procedure, the requirements for hazardous reagents and conditions such as highly toxic methyl chloroformate [19] for the preparation of compound **11** and the dangerous manipulation of hydrogen gas [20] still remained. Therefore, the development of a more efficient and safer synthetic route for the manufacture of pynegabine is highly desirable. Herein, we describe a concise, three-step synthetic route for the preparation of pynegabine at the 10 g scale with easily handling reagents and conditions.

## 2. Results and Discussions

Our proposed new route is shown in Figure 2. The core reaction of this three-step strategy is the palladium-catalyzed Buchwald–Hartwig cross coupling of bromide **17** and commercially available *p*-fluorobenzylamine (**18**). The Buchwald–Hartwig reaction is a highly versatile protocol used to forge aryl C–N bonds [21]. In the present case, this reaction would considerably shorten the synthesis route by replacing the two-step reductive amination (from **13** to **15**) and omitting the nitrogen-installation steps (nitration and hydrogenation from **11** to **13**). Bromide **17** can be prepared from commercially available 2,6-dimethyl-4-bromoaniline (**16**), and the known compound **15** can be transformed to pynegabine by a process similar to that applied in the original route (Figure 1).

### 2.1. Methoxycarbonylation of Aniline ***16***

To avoid the toxic methyl chloroformate (from **10** to pynegabine and from **1** to **11** in Figure 1), we applied an unusual method [22] for the methoxycarbonylation of aniline **16** (Table 1). Treatment of **16** with di-*tert*-butyl dicarbonate (Boc_2_O) and 4-(dimethylamino)pyridine (DMAP) in MeCN afforded isocyanate **19** within 10 min at room temperature. The addition of MeOH to the reaction mixture, followed by heating to reflux, provided carbamate **17** in full conversion. The purification of **17** was simply achieved without column chromatography through acidic aqueous washing to remove DMAP, followed by recrystallization. Using the 0.1 equivalent of DMAP instead of the stoichiometric amount in the literature [22] did not affect the yield (Table 1, entry 2) and the reaction could be scaled up to 20 g scale (Table 1, entry 3). Notably, in this reaction, MeOH should be added immediately after the complete formation of **19** because of the easy oligomerization property of isocyanate [23]. We observed a bulk of precipitate (presumably polyisocyanate) when extending the isocyanation step to 30 min before MeOH addition in a preliminary experiment. Additionally, the use of MeOH instead of MeCN at the beginning of synthesis (Table 1, entry 4) resulted in low conversion, thereby confirming the importance of MeCN as the initial solvent.

### 2.2. Buchwald–Hartwig Cross Coupling of the Compounds ***17*** and ***18***

For the key Buchwald–Hartwig cross coupling between the compounds **17** and **18**, we initially screened the reaction conditions using allylpalladium (II) chloride dimer as the precatalyst and BrettPhos as the ligand [24]. With Cs_2_CO_3_ as the base (Table 2, entries 1–4), ether-type solvent gave the optimal conversion at 95 °C, and a higher yield was obtained with a stronger base (*t*BuONa) in tetrahydrofuran (THF) at a lower temperature (Table 2, entry 5). Further ligand screening (Table 2, entries 6–12) revealed that BippyPhos [25] was most suitable for this coupling reaction (Table 2, entry 12). However, since substrate **18** and product **15** were both amines, the excess **18** which could not be fully consumed need to be removed by column chromatography (Table 2, entry 13). Therefore, using a slight excess of **17** was tested. Under this condition, **18** was completely consumed and pure **15** was successfully obtained through acid–base extraction and recrystallization without column chromatography (Table 2, entry 14). Further optimization of the reaction temperature and time (Table 2, entry 15) showed that a slightly higher temperature (50 °C) gave better yield and reduced the reaction time to a few hours. Scale-up of the reaction to decagram levels maintained the yield level (Table 2, entries 16 and 17).

Interestingly, ^1^H-NMR spectra of compound **15** revealed two sets of peaks (integration ratio: ~2:1) for OMe and amide NH groups in CDCl_3_ (Figure 2a). This might be due to the slow rotation of the hindered carbamate C–N bond [26]. Changing the solvent to d_6_-DMSO altered the integration ratio to ~4:1, confirming its origination from the *cis*-*trans* isomerization of the amide group. The ^13^C-NMR spectra also showed a similar peak pattern (see Appendix A).

### 2.3. Propargylation of Compound ***15***

The third step in pynegabine synthesis involves the propargylation of compound **15**. We reproduced this reaction successfully at a 10 g scale with an 85% yield using a similar procedure as that reported previously (Figure 1b). Additionally, we found that the target pynegabine also showed amide-isomeric peaks in the NMR spectra using CDCl_3_ and d_6_-DMSO as the solvents (Figure 2b).

## 3. Materials and Methods

### 3.1. General Information

Reagents were used as received from commercial suppliers unless otherwise indicated. Reaction solvents (MeCN, MeOH, THF, DMF) were purchased as anhydrous solvents (H_2_O ≤ 50 ppm) and stored with active molecular sieves 3A or 4A under Ar. All solvents for work-up procedures were used as received. Analytical thin-layer chromatography (TLC) was performed using glass TLC plates (silica gel HSGF254 plates, Yantai Huayang New Material Technology Co., Ltd., Yantai, China). Visualization was accomplished with UV light or staining with 5% (*w*/*v*) phosphomolybdic acid/EtOH followed by heating.

Melting points were uncorrected and measured with a Yanaco MP-J3 melting point apparatus. ^1^H- and ^13^C-NMR spectrum were acquired on a Bruker AVANCE III-400 spectrometer. Chemical shifts are indicated in parts per million (ppm) downfield from tetramethylsilane (TMS, δ = 0.00) with residual undeuterated solvent peaks used as internal references for ^1^H-NMR and deuterated solvent peak shifts for ^13^C-NMR. Multiplicities are reported as s (singlet), d (doublet), t (triplet), q (quartet), m (multiplet), br (broad), or combinations of those. Mass spectra were generated using electrospray ionization (ESI) and measured on a Thermo-Fisher Accela liquid chromatography system coupled with an Exactive Plus Orbitrap mass spectrometer.

### 3.2. Synthesis of Methyl (4-Bromo-2,6-dimethylphenyl)carbamate (***17***)

DMAP (1.22 g, 10.0 mmol) and 4-bromo-2,6-dimethylaniline (compound **16**, 20.0 g, 100 mmol) were successively added to a solution of Boc_2_O (30.5 g, 140 mmol) and MeCN (200 mL) in a 500-mL flask at room temperature under an Ar atmosphere. The reaction mixture was stirred for 10 min, followed by the addition of MeOH (50 mL) and reflux at 82 °C for 8 h. After cooling to room temperature, the reaction mixture was concentrated under reduced pressure. CH_2_Cl_2_ (40 mL) was then added to the solid residue, and the organic phase was washed with 1 mol/L hydrochloric acid (40 mL × 3), dried with Na_2_SO_4_, filtered, and concentrated under reduced pressure. The solid residue was then dissolved in CH_2_Cl_2_ (60 mL), and petroleum ether (150 mL) was added dropwise under stirring to induce precipitation. The precipitate was filtered to obtain compound **17** (16.90 g, 65% yield) as a white solid. m.p. 108.0–109.5 °C. ^1^H-NMR (400 MHz, CDCl_3_) δ 7.22 (s, 2H), 5.97 (s, 1H), 3.76 (s, 3H), 2.23 (s, 6H). ^13^C-NMR (101 MHz, CDCl_3_) δ 154.7, 138.0 (2C), 132.7, 131.0 (2C), 120.7, 52.7, 18.2 (2C). HRMS (ESI): *m*/*z* 258.0126 [M+H]^+^ (calcd for C_10_H_13_BrNO_2_^+^: 258.0124).

### 3.3. Synthesis of Methyl (4-(4-Fluorobenzylamino)-2,6-dimethylphenyl)carbamate (***15***)

THF (94 mL) and 4-fluorobenzylamine (compound **18**, 5.26 mL, 5.76 g, 46.0 mmol) were successively added to the mixture of compound **17** (12.5 g, 48.4 mmol), [Pd(allyl)Cl]_2_ (0.443 g, 1.21 mmol), and BippyPhos (2.45 g, 4.83 mmol) in a 350 mL flask at room temperature under an Ar atmosphere. The reaction mixture was stirred for 10 min, followed by the addition of *t*BuONa (1 mol/L in THF, 143 mL, 143 mmol) and stirring at 50 °C for 2 h. After cooling to room temperature, 36% hydrochloric acid (7 mL) was added to adjust the pH to 6, and the mixture was concentrated under reduced pressure. Methyl *tert*-butyl ether (100 mL) was added to the solid residue, and the organic phase was extracted with 0.1 mol/L hydrochloric acid (100 mL × 15). The combined aqueous layer was then adjusted to pH 13 with 1 mol/L aqueous NaOH and extracted with CH_2_Cl_2_ (750 mL × 2). The combined organic layer was then dried with Na_2_SO_4_, filtered, and concentrated under reduced pressure. Petroleum ether was added dropwise to the residue under stirring to induce precipitation. The precipitate was filtered to give compound **15** (10.92 g, 78% yield) as a white solid. m.p. 123.6–124.9 °C. ^1^H-NMR (400 MHz, CDCl_3_) δ 7.31 (dd, *J* = 8.8, 5.6 Hz, 2H), 7.02 (dd, *J* = 8.8, 8.8 Hz, 2H), 6.33 (s, 2H), 5.86 (s, 0.66H), 5.73 (s, 0.34H), 4.25 (s, 2H), 3.94 (s, 1H), 3.74 (s, 2H), 3.68 (s, 1H), 2.16 (s, 6H). ^1^H-NMR (400 MHz, *d*_6_-DMSO) δ 8.22 (s, 0.8H), 7.95 (s, 0.2H), 7.37 (dd, *J* = 8.4, 5.6 Hz, 2H), 7.13 (dd, *J* = 8.8, 8.4 Hz, 2H), 6.26 (s, 2H), 6.06 (t, *J* = 6.0 Hz, 1H), 4.22 (d, *J* = 6.0 Hz, 2H), 3.58 (s, 2.4H), 3.48 (s, 0.6H), 1.99 (s, 6H). ^13^C-NMR (101 MHz, CDCl_3_) δ 162.0 (d, *J* = 245.1 Hz), 156.8 (minor), 155.6 (major), 147.0, 137.2 (2C), 135.1, 129.0 (d, *J* = 7.9 Hz, 2C), 124.3 (minor), 123.9 (major), 115.4 (d, *J* = 21.3 Hz, 2C), 112.3 (major, 2C), 112.0 (minor, 2C), 52.8 (minor), 52.4 (major), 47.6, 18.5 (2C). ^13^C-NMR (101 MHz, *d*_6_-DMSO) δ 161.0 (d, *J* = 241.4 Hz), 155.8 (minor), 155.4 (major), 146.8, 136.5 (d, *J* = 2.9 Hz), 136.0 (major, 2C), 135.8 (minor, 2C), 128.9 (d, *J* = 7.9 Hz, 2C), 124.1 (minor), 123.8 (major), 115.0 (d, *J* = 21.2 Hz, 2C), 111.4 (2C), 51.6 (minor), 51.4 (major), 45.6, 18.3 (2C). HRMS (ESI): *m*/*z* 303.1508 [M + H]^+^ (calcd for C_17_H_20_FN_2_O_2_^+^: 303.1503).

### 3.4. Synthesis of Pynegabine

Diisopropylethylamine (12.1 mL, 8.98 g, 69.5 mmol) and propargyl bromide (3.59 mL, 4.95 g, 41.6 mmol) were successively added to a solution of compound **15** (10.5 g, 34.7 mmol) and DMF (84 mL) in a 500 mL flask at room temperature under an Ar atmosphere. The reaction mixture was then stirred at 60 °C for 14 h and after cooling to room temperature, water (350 mL) was added to induce precipitation. Filtration gave a solid residue that was dried and dissolved in CH_2_Cl_2_ (30 mL). Petroleum ether (90 mL) was added dropwise under stirring to induce precipitation, and the precipitate was filtered to give pynegabine (10.02 g, 85% yield) as a white solid. m.p. 136.2–137.6 °C. ^1^H-NMR (400 MHz, CDCl_3_) δ 7.27 (dd, *J* = 8.8, 5.2 Hz, 2H), 7.01 (dd, *J* = 8.8, 8.8 Hz, 2H), 6.59 (s, 2H), 5.92 (br, 1H), 4.47 (s, 2H), 3.95 (s, 2H), 3.75 (s, 2H), 3.68 (s, 1H), 2.20 (brs, 7H). ^1^H-NMR (400 MHz, *d*_6_-DMSO) δ 8.36 (s, 0.8H), 8.08 (s, 0.2H), 7.33 (dd, *J* = 8.8, 6.0 Hz, 2H), 7.15 (dd, *J* = 8.8, 8.8 Hz, 2H), 6.54 (s, 2H), 4.48 (s, 2H), 4.09 (s, 2H), 3.60 (s, 2.4H), 3.50 (s, 0.6H), 3.14 (s, 1H), 2.06 (s, 6H). ^13^C-NMR (101 MHz, CDCl_3_) δ 162.0 (d, *J* = 244.9 Hz), 156.7 (minor), 155.5 (major), 147.6, 136.9 (2C), 134.0, 128.8 (d, *J* = 7.9 Hz, 2C), 125.2 (minor), 124.9 (major), 115.4 (d, *J* = 21.3 Hz, 2C), 113.9 (major, 2C), 113.6 (minor, 2C), 79.5, 72.3, 54.3, 52.8 (minor), 52.4 (major), 39.7, 18.8 (2C). ^13^C-NMR (101 MHz, *d*6-DMSO) δ 161.2 (d, *J* = 242.1 Hz), 155.7 (minor), 155.2 (major), 146.1, 136.0 (major, 2C), 135.9 (minor, 2C), 134.9 (d, *J* = 2.6 Hz), 129.0 (d, *J* = 8.0 Hz, 2C), 125.8 (minor), 125.5 (major), 115.1 (d, *J* = 21.2 Hz, 2C), 113.2 (2C), 80.5, 74.6, 53.4, 51.7 (minor), 51.5 (major), 40.1, 18.5 (2C). Elemental analysis calcd for C_20_H_21_FN_2_O_2_: C, 70.57; H, 6.22; N, 8.23. Found: C, 69.61; H, 6.09; N, 8.23. HRMS (ESI): *m*/*z* 341.1667 [M + H]^+^ (calcd for C_20_H_22_FN_2_O_2_^+^: 341.1660).

## 4. Conclusions

We successfully developed a concise, three-step route for the decagram-scale synthesis of the AED candidate pynegabine (Figure 3). This route provided a 41% total yield without column chromatographic purification and avoided the original dangerous manipulation of methyl chloroformate and hydrogen gas, thereby making this method more suitable for pynegabine manufacture in the subsequent phases of clinical trials and future marketing.

## 5. Patents

S.X., Y.-J.S., S.-P.Z., Y.-L.G. and S.-C.L. are inventors on patent application 202310142629.9 submitted by the Institute of Materia Medica, Chinese Academy of Medical Sciences, that covers a part of the work of this manuscript.

## Data Availability

The data presented in this study are available in the Materials and Methods section and the Appendix A of this article.

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
