# Peer review of "Three-Step Synthesis of the Antiepileptic Drug Candidate Pynegabine"

_molecules, 2023, doi:10.3390/molecules28134888_

Round 1
Reviewer 1 Report
Below, there is the review of the manuscript (molecules-2446856-peer-review-v1) entitled “Three‐step Synthesis of the Antiepileptic Drug Candidate Pynegabine”
In this paper, the Authors presented a new synthesis method of a potential anti-elliptic drug Pynegatibine.
My comments on the manuscript are as follows:
- In the introduction, the Authors should supplement the literature data on the latest pharmacological findings on pynegabine. The Authors should also provide a broader context for research in the search for more active, safer anti-epileptic compounds with better pharmacological performance as potential drugs than the withdrawn regitabine.
- The Authors did not provide the total yield of the presented synthesis. The Authors gave only the synthesis yields of individual steps. Please provide the yield of the total synthesis. Were the yield values calculated on the basis of a single reaction of every step? or are they the averages of several repetitions of every step? I suggest that the Authors make a table with comparative yields of different methods of pynegabine synthesis (including stages and total synthesis yields).
- Intermediate (isocyanate) 18 (Table 1) has the same number as 4-fluorobenzylamine (Scheme 18, Table 2). Please correct this throughout the manuscript.
- Preparative recipes should be described more precisely, with the solvent amount added (not "small amount of CH2Cl2"). Please, indicate how much methylene chloride and petroleum ether were added
- The preparative recipe shows (3.4. section) that the final product (pynegabine) is raw. The Authors do not report whether they subjected the final product to purification. So, there is no confirmation of its purity. The Authors should perform a combustion analysis to determine purity.
- In the descriptions of 13C-NMR spectra, the Authors give "major" or "minor". Instead, they should provide, for example, assignments of chemical shifts to specific carbon atoms.
- I suggest that the Authors should add a chapter (Conclusions) summarizing the results of the study.
- Please, clearly indicate in the reference list which references are patents.
To summarize. The manuscript should be thoroughly revised and supplemented, and then it can be considered for acceptance for publication.
Reviewer 2 Report
In this communication, Zhang, Xu et al. have reported the development of an intriguing synthetic strategy for producing pynegabine, a drug in phase I clinical trials for treating epilepsy in China. First, they have presented in the introduction, the two procedures used for synthesizing Pynegabine, highlighting their critical points. On that basis, they have developed a three-step synthetic strategy, avoiding for any step, the column chromatography purification procedures. The first step is a carboxylation of the nitrogen atom with t-butyl decarbonate and DMAP (4-dimethylamino pyridine) in acetonitrile. Subsequently, the addition of methanol has given the methoxycarboxlated compound. Furthermore, they have optimized the reaction conditions by varying the quantity of DMAP, obtaining same yields. The second step is the Buchwald–Hartwigd Cross Coupling reaction among the product of the first step and fluoro aniline. The have evaluated the effect of the ligand and temperature. Furthermore, in Fig.2 they reported the spectra of the product vs pynegabine. The third step is the propargylation of the coupling compound.
The paper is well-written, and the results are clear and easy to understand. I suggest the publication in Molecules, only after the authors have checked the following points:
· Scheme 2 and Scheme 3 are too similar. I suggest to modify or scheme 2.
Reviewer 3 Report
The presented manuscript reports on the three‐step route for the decagram‐scale synthesis of pynegabine. The proposed route provided a 41% total yield without column chromatographic purification, which is worthy of note. However, I rate the overall synthetic utility of the proposed solution as moderate. Previous methods of synthesizing this compound were admittedly somewhat longer, but they were of low complexity and much easier for industrial applications. Regarding criticisms of known methods, it can be stated , that reductive amination leading to compounds 8 and 15 (Scheme 1) can be carried out without the separation of intermediate imines, which shortens the number of steps. In addition, the issue of methyl chloroformate toxicity should not be exaggerated. It is as dangerous as almost any acid chloride, which chemistry students can handle very well. On the other hand, the attractiveness of the method proposed by the authors is diminished by the need to use a palladium catalyst and the associated need to remove traces of this toxic metal from the drug product. In addition, copies of NMR spectra and the associated discussion of apparent hindered rotation in the amide bond can be removed, for this is a fairly common phenomenon. It can be mentioned in one sentence. If the authors would like to prove this, it can be easily done using VTNMR in DMSO.
Round 2
Reviewer 1 Report
Since the Authors have made corrections and sufficiently improved the manuscript, I believe that in its present form, it can be accepted for publication.